# Adrenal Crisis Mimicking COVID-19 Encephalopathy in a Teenager with Craniopharyngioma

**DOI:** 10.3390/children9081238

**Published:** 2022-08-17

**Authors:** Tzu-Chien Chien, Mu-Ming Chien, Tsai-Ling Liu, Hsi Chang, Min-Lan Tsai, Sung-Hui Tseng, Wan-Ling Ho, Yi-Yu Su, Hsiu-Chen Lin, Jen-Her Lu, Chia-Yau Chang, Kevin Li-Chun Hsieh, Tai-Tong Wong, James S. Miser, Yen-Lin Liu

**Affiliations:** 1Department of Pediatrics, Taipei Medical University Hospital, Taipei 110, Taiwan; 2Pediatric Brain Tumor Program, Taipei Cancer Center, Taipei Medical University, Taipei 110, Taiwan; 3Department of Pediatrics, School of Medicine, College of Medicine, Taipei Medical University, Taipei 110, Taiwan; 4Department of Physical Medicine and Rehabilitation, Taipei Medical University Hospital, Taipei 110, Taiwan; 5Department of Physical Medicine and Rehabilitation, School of Medicine, College of Medicine, Taipei Medical University, Taipei 110, Taiwan; 6Graduate Institute of Clinical Medicine, College of Medicine, Taipei Medical University, Taipei 110, Taiwan; 7Department of Medical Imaging, Taipei Medical University Hospital, Taipei 110, Taiwan; 8Department of Radiology, School of Medicine, College of Medicine, Taipei Medical University, Taipei 110, Taiwan; 9TMU Research Center of Cancer Translational Medicine, Taipei Medical University, Taipei 110, Taiwan; 10Division of Pediatric Neurosurgery, Department of Neurosurgery, Taipei Medical University Hospital, Taipei 110, Taiwan; 11Neuroscience Research Center, Taipei Medical University Hospital, Taipei 110, Taiwan; 12Taipei Neuroscience Institute, Taipei Medical University, Taipei 110, Taiwan; 13Department of Pediatrics, City of Hope Comprehensive Cancer Center, Duarte, CA 91010, USA; 14Cancer Center, Taipei Medical University Hospital, Taipei 110, Taiwan

**Keywords:** COVID-19, SARS-CoV-2, craniopharyngioma, panhypopituitarism, adrenal insufficiency, seizure

## Abstract

There is an increasing number of reported cases with neurological manifestations of COVID-19 in children. Symptoms include headache, general malaise, ageusia, seizure and alterations in consciousness. The differential diagnosis includes several potentially lethal conditions including encephalopathy, encephalitis, intracranial hemorrhage, thrombosis and adrenal crisis. We report the case of a 17-year-old boy with a positive antigen test of COVID-19 who presented with fever for one day, altered mental status and seizure, subsequently diagnosed with adrenal insufficiency. He had a history of panhypopituitarism secondary to a suprasellar craniopharyngioma treated with surgical resection; he was treated with regular hormone replacement therapy. After prompt administration of intravenous hydrocortisone, his mental status returned to normal within four hours. He recovered without neurologic complications. Adrenal insufficiency can present with neurological manifestations mimicking COVID-19 encephalopathy. Prompt recognition and treatment of adrenal insufficiency, especially in patients with brain tumors, Addison’s disease or those recently treated with corticosteroids, can rapidly improve the clinical condition and prevent long-term consequences.

## 1. Introduction

In March 2020, the World Health Organization officially declared COVID-19 as a pandemic, which has affected a great number of children worldwide [1]. It predominantly presents with respiratory symptoms; however, neurologic manifestations in children have been increasingly reported [2]. Some children have neurologic manifestations as the only presentation of COVID-19 [3]. Most of the neurologic symptoms, occurring in 20% of cases, are non-specific, including headache and ageusia. Importantly, approximately 1% of pediatric cases develop severe neurological complications including encephalopathy, seizures, Guillain–Barré syndrome and intracranial hemorrhage [4,5]. In hospitalized children with acute COVID-19, risk factors of neurological manifestations included older age and preexisting neurological, cardiovascular or metabolic conditions; children with neurological manifestations are more likely to be admitted to the intensive care unit and with longer hospital stays [6].

Adrenal insufficiency, a potentially life-threatening condition, typically presents with nonspecific symptoms including fatigue, nausea, and vomiting, as well as signs of mineralocorticoid deficiency including hypotension, dehydration, and/or electrolyte abnormalities [7]. Adrenal insufficiency can sometimes present with neurologic symptoms including weakness, lethargy, headache and even seizure and coma [8].

Here, we report the occurrence of adrenal insufficiency in a patient with COVID-19 and a history of suprasellar craniopharyngioma presenting with altered mental status and seizure mimicking COVID-19 encephalopathy.

## 2. Case Presentation

A 17-year-old teenage boy presented with high fever and disturbance of consciousness. He had past history of a suprasellar craniopharyngioma, which had been completely resected 3 years prior to presentation (Figure 1). He had had diabetes insipidus and panhypopituitarism since the onset of craniopharyngioma that had persisted postoperatively, for which he received regular follow-up and hormone replacement therapy with oral cortisone acetate (25 mg twice daily), desmopressin, and levothyroxine. He was unvaccinated against COVID-19.

On the morning of presentation, the patient suffered from intermittent high fever. He only took cortisone acetate 25 mg in the morning and was too sick to take the evening dose. A rapid antigen test of COVID-19 was positive at home. A few hours later, he became drowsy and soon developed generalized tonic–clonic seizures for more than 10 min. Upon arrival in the Emergency Department, his vital signs were blood pressure, 103/58 mmHg; heart rate, 108 beats per minute; respiratory rate, 18 times per minute; body temperature, 39 °C. He had obesity based on the Ministry of Health and Welfare of Taiwan; his weight was 78 kg and height was 163 cm, with a body mass index (BMI) of 29.4 kg/m^2^ (96.5 percentile or a Z-score of 1.76, based on Centers for Disease Control and Prevention published BMI reference standards). Physical exam showed normal breath sounds, rapid heart rate, warm extremities and capillary refill time of less than 2 s. Neurological exam showed drowsiness with a Glasgow Coma Scale of E3V3M5, symmetric pupil size (2.5 mm) with light reflex, no Kernig or Brudzinski sign and no drop foot.

COVID-19 quantitative reverse-transcriptase polymerase chain reaction (QRT-PCR) showed positive with a threshold cycle (Ct) value of 14.2, a high viral load in acute phase. Laboratory data revealed increased levels of C-reactive protein (2 mg/dL), procalcitonin (0.5 ng/mL) and interleukin-6 (IL-6; 145.3 pg/mL) (Table 1). Computed tomography showed no space-occupying lesions, no cerebral edema and no intracranial hemorrhage (Figure 2). Lumbar puncture was attempted to collect cerebral spinal fluid but failed.

After initial multidisciplinary consultations, adrenal crisis was the primary diagnosis, with differential diagnosis including acute encephalitis of COVID-19, multi-system inflammatory syndrome in children (MIS-C), and concomitant herpetic encephalitis. An immediate dose of 100 mg of hydrocortisone was loaded intravenously in the ED, followed by 50 mg every six hours (100 mg/m^2^/day). Intravenous acyclovir was also given empirically. The patient was admitted to the Intensive Care Unit for close observation. After a multidisciplinary discussion, a 3-day-course of Remdesivir, 200 mg on the first day and 100 mg on the next two days, was administered. His consciousness and mental status returned normal at 4 h after administration of hydrocortisone. Acyclovir was discontinued because of the rapid response to hydrocortisone and the very low likelihood of herpes simplex virus infection. We gradually tapered the hydrocortisone dose and changed to oral cortisone acetate at the previous physiological dose. The patient was discharged with no neurological sequelae on the fifth day after diagnosis of COVID-19.

## 3. Discussion

### 3.1. COVID-19 in Children with Cancer and Brain Tumor

Our case is an adolescent brain tumor survivor who experienced adrenal insufficiency related to acute COVID-19 infection. Most of the children with malignancies have a mild disease of COVID-19 [9,10]. However, pediatric oncology patients are prone to suffer from worse clinical conditions and more likely to require ICU support when compared to non-oncology patients [11]. Among brain tumors, unlike other oncology diseases, there seems to be no significant difference in mortality from COVID-19. It may be explained by less myelosuppressive brain tumor therapy [12]. Rossella et al. reported a 9-year-old girl with suprasellar non-germinomatous germ cell tumor and hypothalamic-pituitary failure. She was accidently found to be infected with COVID-19 when screened before admission for autologous stem-cell transplantation (ASCT). She was asymptomatic during the course, and she received ASCT one month later [13].

### 3.2. Adrenal Insufficiency in COVID-19 Patients

Adrenal insufficiency is a potentially lethal condition characterized by impaired synthesis and release of adrenocortical hormones. Our patient presented with fever, seizure, and conscious change and was finally diagnosed with adrenal insufficiency according to his clinical response to corticosteroids. Flokas et al. reported a pediatric post-COVID-19 adrenal insufficiency, presenting as MIS-C [14]. An adolescent presented with altered mental status four days after being COVID-19 positive and was eventually diagnosed with Addison’s disease, which was presumed to be triggered by COVID-19 [15]. These cases illustrate that we should keep adrenal insufficiency in mind when approaching a change in consciousness in patients with COVID-19, especially in patients with predisposing factors including a brain tumor, Addison’s disease or recent use of corticosteroids.

### 3.3. Neurological Manifestation of COVID-19 and the Current Management

Evidence of COVID-19 invasion to the central neural system was found in some cases [16,17]. Sejal et al. described a case of COVID-19-related encephalitis in an adolescent, who also had seizures and altered mental status. After a 10-day-course of Remdesivir, his neurologic symptoms improved on the fourth day [18]. Acute fulminant cerebral edema (AFCE) is proposed to be a phenotype of encephalitis in children [19]. Seizure and status epilepticus are two important significant factors of AFCE [20]. Siddharth et al. described a previously healthy 8-year-old girl, presenting with newly onset seizures and encephalopathy, and rapidly progressed to acute fulminant cerebral edema (AFCE) after COVID-19 infection. Despite receiving dexamethasone, intravenous immunoglobulin, Remdesivir, and anakinra, she eventually developed tonsillar herniation and expired [21]. Multisystem inflammation syndrome in children (MIS-C) is an uncommon complication following COVID-19 infection. MIS-C has a similar clinical picture with Kawasaki disease and is potentially life-threatening. A higher incidence of neurological involvement was observed in children diagnosed with MIS-C related to COVID-19 in up to 34% cases [22]. Our case had symptoms that overlapped with MIS-C. He had three system involvements: neurologic, gastrointestinal, and hematologic. Acute phase reactants, including IL-6, CRP, and procalcitonin, were elevated, although he didn’t fully meet the criteria of MIS-C at the time of admission. Adrenal insufficiency was confirmed on the next day with clinical improvements that made the diagnosis of MIS-C unlikely.

### 3.4. Long COVID

Long COVID is a term to describe persisting symptoms after infection with COVID-19. Among children and adolescents, the incidence of long COVID varied widely from 4 to 66% [23]. A Swedish case series reported five pediatric patients of long COVID. Common symptoms include fatigue, dyspnea, chest pain, cognitive impairment, and depression, lasting up to 6-8 months after infected with COVID-19 [24]. Older age, ICU admission, and muscle soreness on admission are significantly related to pediatric long COVID [25]. Long-term follow-up in our case is required. We arranged outpatient department of rehabilitation for this case to detect any signs of long COVID and start an early rehabilitation program.

### 3.5. Limitations

In summary, we report a rare case with a brain tumor experiencing COVID-19 infection and adrenal insufficiency. After prompt hydrocortisone administration, he had a rapid recovery of neurological symptoms. We reinforced to the patient and family about the need to double the dose of cortisone acetate during major illnesses.

Our limitation is that we lack CSF analysis and magnetic resonance imaging (MRI) so we can’t fully rule out encephalitis. As the patient’s consciousness improved rapidly after the administration of steroids and his neurologic examination revealed no remarkable findings, we deferred MRI arrangement and repeating the lumbar puncture. Had there been no clinical improvement, we would have performed a lumbar puncture again and arranged an MRI.

## 4. Conclusions

In a patient with COVID-19 presenting with altered mental status, adrenal insufficiency should always be considered in the differential diagnosis, because adrenal insufficiency can present with neurologic manifestations mimicking COVID-19 encephalopathy. Prompt recognition and treatment of adrenal insufficiency can rapidly improve the clinical condition and prevent long-term consequences. Though our case had no neurologic sequelae one week later, long-term follow-up is needed to evaluate the possibility of long COVID syndrome.

## Figures and Tables

**Figure 1 children-09-01238-f001:**
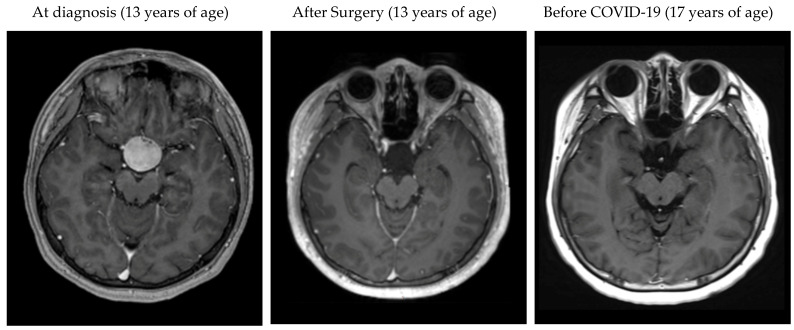
Neuroimaging of the patient’s craniopharyngioma before and after surgical treatment. (**a**) At diagnosis: Magnetic resonance imaging (MRI) at the age of 13 years showed a round, well-circumscribed, vividly enhanced, suprasellar tumor measured 27 × 28 × 29 mm in size. (**b**) After surgery: MRI on the 7th postoperative day showed a gross total resection. (**c**) Most recent follow-up: MRI at 3 months before onset of COVID-19 showed no signs of tumor progression. All images were T1-weighed magnetic resonance imaging with contrast enhancement.

**Figure 2 children-09-01238-f002:**
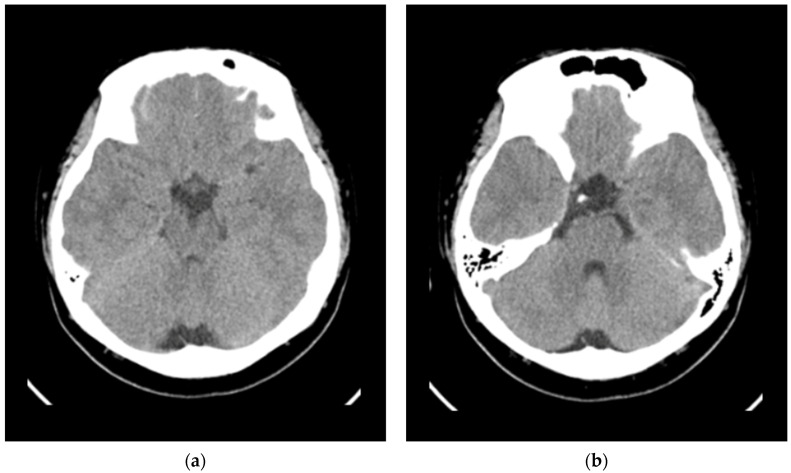
Two different cuts of brain CT revealed no space-occupying lesions, no cerebral edema and no intracranial hemorrhage. (**a**) Level of the midbrain. (**b**) Level of the pons.

**Table 1 children-09-01238-t001:** Laboratory data at presentation.

Test	Our Case	Normal Range
White blood cell	6700/uL	4000–11,000/uL
Hemoglobin	12.8 g/dL	13–17 g/dL
Platelet	211 × 10^3^/uL	130–400 × 10^3^/uL
Neutrophil	59.3%	40–74%
Lymphocyte	21.1%	19–48%
Monocyte	17.6%	2–12%
Eosinophils	1.5%	0–7%
Basophils	0.5%	0–2%
Prothrombin time	15.9 s	11–15 s
aPTT	53.9 s	32–45 s
Fibrinogen	350 mg/dL	200–400 mg/dL
D-dimer	0.73 ug/mL	<0.5 ug/mL
FDP	<4 ug/mL	<5 ug/mL
C-reactive protein	2.01 mg/dL	<0.5 mg/dL
Procalcitonin	0.51 ng/mL	<0.046 ng/ml
Interleukin-6	145.3 pg/mL	<7 pg/mL
Blood urea nitrogen	13 mg/dL	6–20 mg/dL
Creatinine	0.8 mg/dL	0.7–1.2 mg/dL
AST	88 U/L	<40 U/L
ALT	49 U/L	<41 U/L
Direct bilirubin	0.4 mg/dL	0–0.3 mg/dL
Total bilirubin	0.9 mg/dL	0–1.2 mg/dL
Creatinine kinase	44 U/L	20–200 U/L
Creatinine kinase-MB	12 U/L	<25 U/L
Troponin T	0.006 ng/mL	0–0.014 ng/mL
Glucose	149 mg/dL	80–140 mg/dL
Ammonia	28 ug/dL	27–102 ug/dL
Lactate	6.1 mg/dL	4.5–19.8 mg/dL
Na	133 mEg/L	136–145 mEg/L
K	3.4 mEg/L	3.5–5.1 mEg/L
Ca	8.6 mg/dL	8.6–10.2 mg/dL
Mg	1.9 mg/dL	1.6–2.6 mg/dL
P	4.7 mg/dL	2.7–4.5 mg/dL
Cortisol (PM) *	0.05 ug/dL	7–10 AM: 6.2–19.4 ug/dL 4–8 PM: 2.3–11.9 ug/dL
ACTH *	10.3 pg/mL	7.9–47.1 pg/mL
T3 *	152 ng/dL	80–200 ng/dL
Free T4 *	1.7 ng/dL	0.93–1.7 ng/dL
TSH *	<0.01 ulU/mL	0.27–4.2 ulU/mL

* The endocrine function test results were available at 12 h after initial evaluation. Abbreviations: ACTH: adrenocorticotropic hormone; aPTT: activated partial thromboplastin time; AST: aspartate aminotransferase; ALT: alanine transaminase; TSH: thyroid stimulating hormone.

## Data Availability

Not applicable.

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
