# Peer review of "Adrenal Crisis Mimicking COVID-19 Encephalopathy in a Teenager with Craniopharyngioma"

_children, 2022, doi:10.3390/children9081238_

Round 1
Reviewer 1 Report
This is a well written case presentation of coincident COVID-19 infection and panhypopituitarism secondary to resection of a craniopharyngioma. Although to me as a pediatric neurosurgeon I would think it would be second nature for any pediatrician or pediatric emergency room physician to administer stress doses of hydrocortisone to any patient with panhypopituitarism, perhaps this knowledge is not widely known and this manuscript would serve as an educational effort
Reviewer 2 Report
The study presents an interesting case of adrenal insufficiency during COVID-19 in a patient after surgical treatment of craniopharyngioma.
The case is well described and discussed including the available publications.
I have one minor comment regarding BMI Z-score that could be added to a patient case description.
Reviewer 3 Report
The report from Chien et al. describes a case of COVID-19 leading to adrenal insufficiency in a patient previously treated with surgical resection of a craniopharyngioma. The manuscript is well written and clearly describes the case. It points out the importance of considering adrenal insufficiency as a diagnose when patients present with neurologocal manifestations and COVID-19.
I only have a few minor points:
Line 65: "symptoms" is missing after nonspecific
Line 133: "of" should be removed after myelosuppressive
Line 151: it should be: "in an adolescent"
Line 170: it should be "infection" instead of "infected"
Line 171: it should be "adolescents" instead of "adolescence"
